# Optimization of Gefitinib-Loaded Nanostructured Lipid Carrier as a Biomedical Tool in the Treatment of Metastatic Lung Cancer

**DOI:** 10.3390/molecules28010448

**Published:** 2023-01-03

**Authors:** Abdelrahman Y. Sherif, Gamaleldin I. Harisa, Ahmad A. Shahba, Fars K. Alanazi, Wajhul Qamar

**Affiliations:** 1Kayyali Chair for Pharmaceutical Industry, College of Pharmacy, King Saud University, Riyadh 11451, Saudi Arabia; 2Department of Pharmaceutics, College of Pharmacy, King Saud University, Riyadh 11451, Saudi Arabia; 3Department of Biochemistry and Molecular Biology, College of Pharmacy, Al-Azhar University, Nasr City 11884, Cairo, Egypt; 4Department of Pharmacology and Toxicology, College of Pharmacy, King Saud University, Riyadh 11451, Saudi Arabia

**Keywords:** gefitinib, NLC, LCFA, DOE, cytotoxicity, lung cancer

## Abstract

Gefitinib (GEF) is utilized in clinical settings for the treatment of metastatic lung cancer. However, premature drug release from nanoparticles in vivo increases the exposure of systemic organs to GEF. Herein, nanostructured lipid carriers (NLC) were utilized not only to avoid premature drug release but also due to their inherent lymphatic tropism. Therefore, the present study aimed to develop a GEF-NLC as a lymphatic drug delivery system with low drug release. Design of experiments was utilized to develop a stable GEF-NLC as a lymphatic drug delivery system for the treatment of metastatic lung cancer. The in vitro drug release of GEF from the prepared GEF-NLC formulations was studied to select the optimum formulation. MTT assay was utilized to study the cytotoxic activity of GEF-NLC compared to free GEF. The optimized GEF-NLC formulation showed favorable physicochemical properties: <300 nm PS, <0.2 PDI, <−20 ZP values with >90% entrapment efficiency. Interestingly, the prepared formulation was able to retain GEF with only ≈57% drug release within 24 h. Furthermore, GEF-NLC reduced the sudden exposure of cultured cells to GEF and produced the required cytotoxic effect after 48 and 72 h incubation time. Consequently, optimized formulation offers a promising approach to improve GEF’s therapeutic outcomes with reduced systemic toxicity in treating metastatic lung cancer.

## 1. Introduction

Tyrosine kinases play an important role in cell proliferation by affecting signaling pathways, DNA repair, and programmed cell death [1]. Therefore, tyrosine kinase inhibitors are used during the treatment of different types of cancer [2]. Among them, gefitinib (GEF, abbreviations listed in Appendix A) is approved for the treatment of non-small cell lung cancer [3]. It belongs to class II according to biopharmaceutical classification systems, with a Log *p* value of (3.2); it is highly hydrophobic with low solubility [2]. GEF has an oral bioavailability of around 44%, which delays its onset of action and requires increasing the dose [4]. This results in several GEF-related undesirable side effects such as hepatic dysfunction, anorexia, stomatitis, vomiting, diarrhea, and nausea [3]. Moreover, GEF resistance limits its effective clinical application [1].

Nanotechnology is proposed as an effective approach to resolve the therapeutic problems of anticancer agents by increasing drug solubility and bioavailability [3]. The encapsulation of therapeutic agents within nanocarriers enhances drug biodistribution to the targeted cells [5]. This could be achieved during the treatment of cancer where nanoparticles are inherently distributed to tumor tissues based on enhanced permeability and retention phenomena [6]. Upon oral administration, the lipid-based nanocarriers, consisting of long-chain fatty acids, predominantly follow the track of the chylomicron absorption pathway through the lymphatic system [7,8]. The administrated lipids stimulate bile salt secretion, which encourages the formulation of colloidal emulsion within the intestinal lumen and enhances drug bioavailability [9]. Furthermore, negatively charged nanoparticles are susceptible to lymphatic delivery via M cells [10], which evades first-pass metabolism and increases drug bioavailability [7].

Solid lipid nanoparticles (SLNs), nanostructure lipid carriers (NLCs), liposomes, self-nano-emulsifying drug delivery systems, emulsions, and other lipid-based drug delivery systems have been utilized to enhance drug dissolution, bioavailability, and increase drug absorption through the lymphatic system [11,12]. Among them, NLC formulations have several advantages, including sustained drug release, high drug entrapment efficiency, the possibility of large-scale production, and superior stability compared to other lipid-based formulations. The internalization of NLCs into cancer cells is quicker compared with other lipid-based formulations [13]. NLCs are composed of liquid oils and solid lipids in the internal core, surrounded by surfactants as a stabilizer. The presence of liquid oils decreases the degree of solid lipid crystallinity, which avoids drug expulsion and instability during storage [14,15]. NLC formulations promote drug oral absorption via selective uptake through the enterocytes or the payer’s patch [16]. It has been reported that tyrosine kinase inhibitors loaded into NLCs improves their therapeutic efficacy and organ drug targeting and reduces the side effects [17].

The present work aimed to develop NLCs as an effective drug delivery system that could enhance GEF therapeutic outcomes in lung cancer management. In the current study, stearic acid (SA) was used as the solid lipid during the fabrication of NLC formulations. Three different types of liquid oils (oleic acid, glycerol monolinoleate, and soybean) were investigated; namely, long-chain fatty acids (LCFA), long-chain monoglyceride (LCM), and long-chain triglyceride (LCT), respectively. The impact of different liquid oils (LCFA, LCM, and LCT) at different solid lipid: liquid oil ratios (SL/LO) on the physicochemical properties of plain NLC was studied using Design of Experiments (DOE). The optimized plain NLC formulations were loaded with GEF (GEF-NLC) and characterized in terms of particle size (PS), polydispersity index (PDI), zeta potential (ZP), drug loading, entrapment efficiency, and stability. Moreover, the cytotoxic activity of GEF-NLC_(LCFA)_ and free GEF were investigated based on the viability of A549 cells as a model for lung cancer (using an MTT assay).

## 2. Results and Discussion

### 2.1. Solubility of GEF in Liquid Oils

GEF solubility in LCFA, LCM, and LCT was 207.54 ± 14.63, 42.42 ± 5.71, and 0.41 ± 0.02 mg/g, respectively. The detected maximum solubility of GEFs in LCFA was attributed to the acid microenvironment generated with the free carboxylic group. This resulted in the protonation of the amino group of weak basic GEFs, which enhanced its solubilization [9]. In addition, GEFs exhibited higher solubility in LCM compared with LCT as a result of the self-emulsification properties of the former [8]. The obtained results were in alignment with the findings shown by Dhairyasheel et al., who reported that GEF solubility depends on the type of liquid oils. Maximum GEF solubility was detected in LCFA, while it was minimum in the case of LCT [18]. Likewise, Shahba et al. studied the solubility of a weakly basic drug in SNEDDS formulations. The results revealed that maximum drug solubility was achieved in SNEDDS formulations containing LCFA compared to their ester counterparts [19].

### 2.2. Effect of Independent Variables on the Responses

The prepared formulations were characterized in terms of physicochemical properties, including PS, PDI, and ZP, as shown in Table 1. DOE software was utilized to statistically analyze the effect of independent variables on each response separately based on different mathematical models (linear, 2FI, Cubic, and Quadratic). The design type and model were selected based on a step-by-step design wizard (Design-Expert^®^ version13, Stat-Ease Inc., Minneapolis, MN, USA). Table 2 shows the selected models for each response based on the ANOVA analysis with a high correlation coefficient, high F-value, non-significant lack of fit, high adjusted and predicted R2 (difference < 0.2), and high adequate precision. The effect of the two independent variables on each response is discussed separately.

#### 2.2.1. PS

PS of the prepared plain NLC formulations ranged from 189.6 to 399.1 nm (Table 1). Figure 1A shows the effect of both independent variables on the PS of plain-NLC. Increasing SL/LO for all types of liquid oils resulted in a significant increase (*p* < 0.05) of plain-NLC PS (Table 3 and Figure 2). Additionally, liquid oil type showed a significant effect (<0.05) on plain-NLC PS at the same SL/LO. According to the type of liquid oil, the droplet size of NLCs were arranged in the following order: LCFA < LCT < LCM, providing that the SL/LO was constantly maintained (Table 3 and Figure 2). The PS of the plain NLC for each liquid oil (LCFA, LCM, and LCT) could be calculated from the final equations (in terms of actual components), Equations (1)–(3), respectively:PS (LCFA) = 192.16 + (15.58 × SL/LO)(1)
PS (LCM) = 197.33 + (79.44 × SL/LO)(2)
PS (LCT) = 203.59 + (32.50437 × SL/LO)(3)

The intestinal transport of nanoparticles predominantly depends on their PS. Smaller particles are mostly uptaken via enterocytes through receptor-mediated endocytosis or phagocytosis processes [20]. In addition, nanoparticles less than 500 nm in size could be delivered to the lymphatic system through M cells [10]. The decrease in PS resulting from the decreasing SL/LO could be attributed to a reduction in the viscosity of the formulation. Therefore, the distribution of energy within the media during the production process was enhanced. In alignment with the obtained results, different research has shown that decreasing SL/LO resulted in the decreased PS of prepared NLC formulations [16,21,22]. The increased PS of plain-NLC-containing LCT compared to LCFA could be attributed to increasing the degree of FAs esterification. In accordance with our findings, other studies demonstrated that the replacement of LCFA with LCT significantly increases PS [23,24].

#### 2.2.2. PDI

All the prepared formulations were homogenously distributed with a PDI value of less than 0.4. The PDI of the prepared NLC formulations ranged from 0.128 to 0.162, 0.258 to 0.349, and 0.175 to 0.234 for SA/LCFA, SA/LCM, and SA/LCT lipid core as shown in Table 1. Figure 1B shows the effect of both independent variables on the PDI of plain-NLC. Increasing SL/LO for all types of liquid oils resulted in a significant increase (*p* < 0.05) of plain-NLC PDI (Table 3 and Figure 3). Additionally, liquid oil type showed a significant effect (<0.05) on plain-NLC PDI value at the same SL/LO. According to the type of liquid oil, the PDI value of plain NLC was arranged in the following order: LCFA < LCT < LCM, providing that the SL/LO is maintained constant (Table 3 and Figure 3). The final equations (in terms of actual components), Equations (4)–(6), could be utilized to predict actual PDI for plain-NLC consisting of liquid oil (LCFA, LCM, and LCT, respectively):PDI (LCFA) = 0.118 + (0.0158 × SL/LO)(4)
PDI (LCM) = 0.251 + (0.0158 × SL/LO)(5)
PDI (LCT) = 0.173 + (0.0158 × SL/LO)(6)

Nanoparticles with PDI values of less than 0.3 are considered homogeneously distributed in the measured PS value [9]. The decrease in PDI value resulting from the decreasing SL/LO could be attributed to a reduction in the viscosity of the formulation. This allows homogenous distribution of energy and produces plain NLC with a low PDI value [9,25].

#### 2.2.3. ZP

Table 1 showed that the ZP values for all plain NLC formulations ranged from −21.8 to −31.7 mV. Figure 1C shows the effect of both independent variables on the ZP value of plain-NLC. However, SL/LO showed an insignificant effect (*p* > 0.05) on the ZP of plain NLC (Table 3 and Figure 4). On the other hand, the liquid oil type showed a significant effect (*p* < 0.05) on ZP at the same SL/LO. The NLC ZP was arranged in the following order: LCFA < LCM < LCT, providing that the SL/LO was constantly maintained (Table 3 and Figure 4). The ZP value of any plain NLC for any liquid oil (LCFA, LCM, and LCT) could be obtained from the final equations, (in terms of actual components), Equations (7)–(9), respectively.
ZP (LCFA) = −30.71 + (0.13 × SL/LO)(7)
ZP (LCM) = −29.98 + (0.13 × SL/LO)(8)
ZP (LCT) = −23.07 + (0.13 × SL/LO)(9)

The surface charge of nanoparticles plays a crucial role in the lymphatic delivery of nanoparticles. It was found that nanoparticles with neutral or negative ZP values are more susceptible to lymphatic uptake through M cells [10]. The surface charge of nanoparticles is predominantly affected by the type of lipid esterification. It was found that increasing the degree of glycerol esterification resulted in a significant increase in ZP value. This is attributed to the negative charge produced by the free carboxylic group in the free FA [26]. The present results are in agreement with various studies that showed that NLC prepared from LCFA has a ZP value less than LCM and LCT [27,28]. This is attributed to the presence of acidic components that produced nanoparticles with lower ZP values. This resulted from the dissociation of acidic groups and produced a more negative value on the surface [29].

### 2.3. Stability of Plain NLC Formulations

The prepared plain NLC formulations were stored in the refrigerator at 4 °C for 3 months to study their stability. Table 1, response Y_4_, shows the aggregation of the prepared plain NLC formulations. Plain NLC formulations comprising LCM showed remarkable particle aggregation within 1–2 days. Formulations comprising LCFA showed particle aggregation within the first week, except for formulations (codes 4, 5, 7, and 12), which contained a high SL/LO (>1.68) (Table 1). The instability of plain NLCs consisting of low SL/LO could be attributed to the presence of a free carboxylic group in the liquid state for LCFA. This could alter the interfacial properties of plain NLCs and encourage particle aggregation [30]. Therefore, increasing the SL/LO could result in decreased mobility of the plain NLC core, which enhances its stability. On the contrary, all formulations comprising LCT showed PS that was maintained below 400 nm within the 90-day storage time. This could be attributed to the conjugation of fatty acids with glycerol, which decreases its activity on the interfacial and surface properties.

### 2.4. Selection of the Optimum Formulation and Validation of DOE

DOE was used to select the formulations with desirable physicochemical properties. Formulations containing LCM were excluded from the suggested formulation to avoid instability during storage. Regarding LCFA, plain NLC formulations with low SL/LO were unstable, and remarkable particle aggregation was observed. Therefore, during the selection of the optimized formulation containing LCFA, the criteria were as follows: SL/LO (1.68–3, maximize, and high priority), type of liquid oil (LCFA), PS, PDI, and ZP (minimize). Even though low SL/LO produces smaller particles, high priority to high SL/LO was selected to enhance formulation stability. In the case of LCT, it was observed that formulations with a low SL/LO of 0.33 showed remarkable particle aggregation after GEF loading, which indicated drug expulsion. This is attributed to the presence of GEF in SA as a result of low solubility in LCT. Therefore, the selection criteria of LCT formulation were chosen to maximize the SL/LO ratio as follows: SL/LO (maximize, and higher priority), type of liquid oil (LCT), PS, PDI, and ZP (minimize). Both GEF-NLC_(LCFA)_ and GEF-NLC_(LCT)_ with 3 SL/LO were suggested by DOE and subjected to further evaluation. However, all the optimized formulations (with 3 SL/LO) showed acceptable PS (<300 nm), even after drug loading (no drug explosion occurred). Accordingly, the decision was made based on balancing between different formulation attributes; in particular, PS and stability. Table 4 and Table 5 show the results for the plain NLC_(LCFA)_ and plain NLC_(LCT)_ optimized formulations, respectively. The actual measurements of both formulations were close to the predicted values of responses and fell within 95% prediction intervals.

### 2.5. Effect of Drug Loading

The physicochemical properties of plain NLCs (LCFA and LCT) and GEF-NLC (LCFA and LCT) are shown in Figure 5. It was found that the PS of GEF-NLC_(LCFA)_ was not significantly (*p* > 0.05) changed upon drug loading, while it was significantly (*p* < 0.05) increased upon GEF loading into plain NLC (LCT). Similarly, ZP showed insignificant change (*p* > 0.05) upon GEF loading into plain NLC (LCFA), while it significantly (*p* < 0.05) increased from −22 to −17 upon GEF loading into plain NLC (LCT). In addition, both formulations showed high drug content above 2 mg/mL, with remarkable entrapment efficiency above 90% (Table 6).

The insignificant increase in the PS of plain NLC (LCFA) after the incorporation of GEF could be attributed to the solubilization of GEF in LCFA. However, the PS of GEF-NLC _(LCT)_ was significantly larger than the corresponding plain-NLC_(LCT)_. This could be attributed to the low solubility of GEF in LCT and its presence in SA. The presence of GEF in solid lipids instead of liquid oil resulted in the disruption of SA crystallization. In a similar context, Sherif et al. found that the incorporation of a drug within a solid lipid resulted in a disruption in SA crystallinity [2]. Regarding ZP, the incorporation of GEF resulted in increased ZP value in the case of GEF-NLC_(LCT)_. This resulted from the neutralization effect produced by positively charged GEF on the negative charge of SA. These results are concurrent with Shahba et al., who found that loading of the weakly basic drug in a lipid-based formulation containing LCFA increased ZP value [31].

### 2.6. PXRD

Visualization of an internal crystalline pattern of a lipid core is very difficult. Therefore, a solidified film of processed SA, processed SA: LCFA (3:1), and processed SA: LCT (3:1) was prepared to resemble SL/LO, which was used to prepare GEF-NLC_(LCFA)_ and GEF-NLC_(LCT)_, respectively. The lipids were melted and left to cool to study the impact of liquid oil on SA crystallinity. Figure 6A–C show that both liquid oils were able to disrupt SA crystallinity while it was remarkable with LCT. The effect of both liquid oils on the crystal morphology was further examined following crushing and mild graining of solidified lipids to obtain fine crystals for SEM examination. The SEM study suggests that the processed SA has a crystalline structure, while the crystallinity was reduced after the addition of LCFA and LCT (Figure 6D–F). Moreover, the degree of crystallinity was reduced in the case of SA: LCT (3:1) when compared with SA: LCFA (3:1).

For further examination, the prepared crushed films of processed SA, processed SA: LCFA (3:1), and processed SA: LCT (3:1) were subjected to PXRD as shown in Figure 7A. PXRD of processed SA shows a high-intensity peak at 6.8, 21.7, 24.3, and 38.1°, along with moderate-intensity peaks at 4.5, 11.0, 11.2, 44.3, and 77.5°. The processed SA: LCFA (3:1) and processed SA: LCT (3:1) were prepared to resemble the SL/LO that is used to prepare GEF-NLC_(LCFA)_ and GEF-NLC _(LCT)_, respectively. The intensity of peaks for SA: LCFA (3:1) at 4.5, 6.8, 11.0, 11.2, and 24.3° was reduced, while peaks at 38.1, 44.3, and 77.5° were slightly increased with no effect on the peak at 21.6°, although a similar observation was detected with SA: LCT (3:1) at the peak range 4.5°–24.3°, while the peaks at 38.1, 44.3, and 77.5° completely disappeared. It is clear from the crystalline pattern that LCT was able to disturb SA crystallinity when compared with LCFA. This could be attributed to the lower melting point of LCT (−16 °C) when compared with LCFA (13–16 °C).

The present findings are in accord with the observation of Galvao et al., who studied the effect of liquid oil on three different solid lipids, including SA. The reported PXRD revealed that liquid oil incorporation resulted in decreased crystallinity of SA [32]. Likewise, Lin et al. prepared three different NLC formulations consisting of Antarctic krill oil as the liquid oil and three types of solid lipids. The degree of all solid lipid crystallinity in NLC formulations was significantly reduced compared to its pure form [33]. Additionally, Das et al. found that the incorporation of liquid oil with solid lipids resulted in a significant reduction in the crystalline pattern of solid lipids [34].

The crystallinity of GEF, processed SA, GEF-NLC_(LCFA)_, and GEF-NLC_(LCT)_ are represented in Figure 7B to evaluate the degree of GEF and SA crystallinity within GEF-NLC formulations. PXRD of GEF shows high-intensity peaks at 38.1° and 44.3°, in addition to multiple peaks with moderate intensity at 19.4, 24.2, 26.4, and 77.5°. A PXRD graph of both NLC (LCFA) and NLC (LCT) showed peaks at 6.7, 21.7, and 24.3° with high intensity, and 4.5 and 11.1° with moderate intensity. Most importantly, the GEF characteristic peaks at 38.1 and 44.3° completely disappeared in the prepared GEF-NLC formulations. This indicates the presence of the drug in the amorphous state within NLC formulations. In harmony with the obtained results, various studies showed that the incorporation of lipophilic drugs in NLC resulted in a significant reduction in drug crystallinity degree [35,36].

### 2.7. In Vitro Release

Figure 8 shows the in vitro release profile of GEF from GEF-NLC_(LCFA)_ and GEF-NLC_(LCT)_ formulations. Burst drug release was observed from GEF-NLC_(LCT)_, where about 70% of the drug was released within the first two hours. In contrast, GEF-NLC_(LCFA)_ showed a gradual drug release, where only 20% of the drug was released within the first two hours. Moreover, a sustained drug release was observed where only about 50% of GEF was released from GEF-NLC_(LCFA)_ within 24 h. It is worth mentioning that the dissolution efficiency of GEF-NLC_(LCFA)_ was significantly (*p* < 0.05) lower than that of GEF-NLC_(LCT)_.

Based on the current solubility study, LCT shows limited GEF solubility. Hence, the amount of LCT, present in GEF-NLC_(LCT)_, was not sufficient to dissolve 5% of the total drug amount. Therefore, as shown in Figure 9, GEF was expected to be predominately present in the solid lipid (SA—a crystalline moiety) rather than liquid oil in the case of GEF-NLC_(LCT)_. Moreover, our PXRD study confirmed that the addition of LCT to SA significantly disrupted its crystallinity and arrangement. Therefore, the drug was easily released from the lipid core of GEF-NLC_(LCT)_. On the other hand, LCFA showed high GEF solubility. Hence, GEF is expected to be distributed in both liquid oil (LCFA) and solid lipid (SA) within GEF-NLC_(LCFA)_ formulation (Figure 9). Moreover, the PXRD study showed that LCFA was less able to disrupt SA crystallinity compared to LCT. Therefore, the observed sustained release in the case of GEF-NLC_(LCFA)_ could be explained by the partial entrapment of GEF within LCFA.

In alignment with this hypothesis, previous studies evaluated drug release from NLCs compared to SLNs (that are free of liquid oil). Zhang et al. and Thatipamula et al. reported that NLCs showed a lower rate of drug release compared to SLNs. Both studies utilized solid lipids and liquid oils with high drug solubility [37,38]. Moreover, in our previous work, GEF-SLN formulation contained a solid lipid only where GEF was expected to be homogenously distributed within SA (Figure 9). An in vitro release study revealed that about 74% of the drug was released within 24 h [2]. Herein, about 50% GEF was released within 24 h from GEF-NLC_(LCFA)_. These findings indicate that the drug is easily released from solidified lipids compared to liquid oils. Moreover, burst drug release was not attained with the GEF-NLC_(LCFA)_ formulation even with the addition of LCFA, which disrupts SA crystallinity. This could be attributed to the superior effect of liquid oil incorporation on drug solubilization within GEF-NLC_(LCFA)_. It can be concluded that the addition of liquid oil not only increases the stability of SLNs but also determines the drug release behavior of drug-loaded NLCs. Finally, it is expected that GEF-NLC_(LCFA)_ could be an attractive option for lymphatic delivery by nanoparticle uptake either via enterocytes or M cells during the endocytosis process [27].

### 2.8. Stability of GEF-NLC Formulations

The PS of GEF-NLC_(LCFA)_ and GEF-NLC_(LCT)_ significantly (*p* < 0.05) increased from 260.6 and 282.1 to 305.8 and 458.7, respectively (Figure 10). However, the droplet size of GEF-NLC_(LCFA)_ was significantly (*p* < 0.05) lower than that of GEF-NLC_(LCT)_, irrespective of storage time. Most importantly, the droplet size of GEF-NLC_(LCFA)_ remained <310 nm for up to 90 days. Finally, the ZP value was negative during storage with no significant difference (*p* > 0.05) in its value.

The significant increase in the PS of GEF-NLC_(LCT)_ could be attributed to the predominant presence of GEF in solid lipids instead of liquid oil. This resulted in drug expulsion from solid lipids owing to reported SA recrystallization during storage [2]. This is in accordance with a previously reported study where the deformation of nanoparticles resulted in particle aggregation [30]. However, the droplet size of GEF-NLC_(LCFA)_ was significantly (*p* < 0.05) lower than that of GEF-NLC_(LCT)_, irrespective of storage time. Most importantly, the droplet size of GEF-NLC_(LCFA)_ remained < 310 nm for up to 90 days. This could be attributed to the solubilization of the drug in liquid lipids or the migration of the drug during solid lipid recrystallization to liquid oil. In alignment with the obtained results, Zhang et al. found that the addition of liquid oil resulted in a significant improvement in nanoparticle stability. The author referred this to as the reduction in drug expulsion from NLCs compared to SLNs [37]. Therefore, it can be concluded that using liquid oil to formulate NLCs with higher drug solubility could enhance the pharmaceutical stability of the prepared formulation. Therefore, GEF-NLC_(LCFA)_ was selected as the optimized formulation based on in vitro release and stability studies and then subjected to a cytotoxicity study.

### 2.9. In Vitro Cytotoxicity

The MTT assay was utilized to study the cytotoxic activity of plain NLC_(LCFA)_, pure GEF, and GEF-NLC_(LCFA)_ on the growth of lung cancer cell lines. Figure 11A–C show the effect of formulations at different concentrations on the cell viability following 24, 48, and 72 h incubation time, respectively. It was found that pure GEF and GEF-NLC exhibited a cytotoxic effect in a strong concentration-dependent manner. In addition, the IC_50_ of pure GEF was 11.16, 3.54, and 2.08 μg/mL following 24, 48, and 72 h incubation, respectively (Figure 11). However, the IC_50_ of GEF-NLC was 15.05, 4.35, and 2.65 μg/mL following 24, 48, and 72 h incubation, respectively (Figure 12). Both formulations showed a significant decrease in IC_50_ upon increasing the incubation time. However, pure GEF showed a significantly lower (*p* < 0.05) IC_50_ compared to GEF-NLC, only at 24 h incubation (Figure 12). Interestingly, it showed no significant difference (*p* > 0.05) from GEF-NLC at subsequent incubation intervals: 48 and 72 h. GEF-NLC_(LCFA)_ exhibited a gradual increase in cytotoxic activity with time (as represented by a gradual decrease in IC_50_).

In the present study, the in vitro release study showed that GEF-NLC_(LCFA)_ exhibited a sustained release profile, and this delays its cytotoxic activity. In this context, the cytotoxic activity produced by GEF-NLC_(LCFA)_ significantly increased following 48 and 72 h incubation time. The current results are in accordance with previous studies, which demonstrated that lipid-based formulation decreases exposure of the cultured cells to cytotoxic agents following 24 h incubation time. This feature could avoid premature drug release before it reaches cancer cells [39,40]. Moreover, following 48 h incubation, GEF-NLC_(LCFA)_ had an IC_50_ value equivalent to pure GEF with no significant difference. This is in agreement with the sustained release profile that was observed in the in vitro drug release study. This avoids drug release within GIT until the GEF-NLC_(LCFA)_ formulation reaches cancer cells. Moreover, the incorporation of lipid components within the prepared nanoparticles ensures predominant cellular uptake via cancer cells with minimal systemic toxicity [17]. This is in agreement with the previously reported studies which demonstrate that lipid-based formulation is subjected to cancer cell uptake via overexpressed receptors [13,41]. Moreover, a further bio-distribution study showed that drug-loaded lipid-based formulation comprising SA enhanced drug deposition in lung tissue following oral administration [42]. Collectively, the prepared formulation is not only expected to reduce systemic toxicity but also increase GEF distribution to lung tissue, as well as the lymphatic system, during the treatment of metastatic lung cancer. Further in vivo studies are still required to address this issue.

## 3. Materials and Methods

### 3.1. Materials

Gefitinib (GEF) was purchased from Beijing Mesochem Technology Co., Ltd., (Beijing, China). Pluronic-F68 was purchased from Sigma Aldrich, (St. Louis, MO, USA). Stearic acid (SA) was purchased from BDH, (Poole, UK). Oleic acid (LCFA) was obtained from Avonchem, (Cheshire, UK). Maisine 35-1 (LCM) was purchased from Gattefossé, Saint Priest, France. Soybean (LCT) was generously provided by John L. Seaton & Co., Ltd., Croda International Plc., (East Yorkshire, UK).

### 3.2. Solubility Study of GEF in Liquid Oils

The solubility of GEF was measured as previously described by Pandey et al. with minor modification. Briefly, an excess amount of GEF was placed in a screw-capped glass vial containing around 1 gm liquid lipid (LCFA or LCM or LCT) and magnetically stirred for 72 h at room temperature. At the end of the experiment, the mixture was centrifuged for 10 min at 15,000 rpm and drug concentration in the supernatant was measured using the developed UPLC method. The solubility of GEF in each oil was measured in triplicate [36].

### 3.3. Design of Experiments (DOE)

A randomized response surface study with a quadratic I-optimal model (Design-Expert^®^ version 13, Stat-Ease Inc., Minneapolis, MN, USA) was used to analyze the effect of independent variables on the designated response [43]. The study involved the assessment of the two independent variables—solid lipid: liquid oil ratio (SL/LO) and type of liquid oil—in terms of their impact on the physicochemical properties of the prepared plain NLC. Response surface methodology (RSM) was used to investigate the effect of the independent variables (X_1_ and X_2_) on a range of dependent variables, including PS (nm), PDI, ZP (mV), and aggregation upon storage, designated as Y_1_, Y_2_, Y_3_, and Y_4_, respectively. Appropriate models were selected by comparing *p* values and coefficient of determination (R^2^) values. The range of each variable was selected as follows: (SL/LO = 0.33–3) and (type of liquid oil = LCFA, LCM, and LCT). Seventeen plain NLC formulations were prepared as suggested by the DOE (Table 1). The correlation of factors with response variables was then fitted into different mathematical models (linear, quadratic, cubic, or special cubic) [44]. Analysis of variance (ANOVA) was applied to determine the significance of each design model, independent variables, and their interactions [45]. For each response, the optimum model was selected based on whether it showed a high correlation coefficient, a high F-value, a non-significant lack of fit, high adjusted and predicted R2 (difference < 0.2), and high adequate precision [46,47]. Response surfaces were constructed using the obtained equations to aid in the selection of the optimized formulation based on PS, PDI, and ZP. Response surface plots were generated to visualize the simultaneous effect of each variable on each response parameter. Afterward, a desirability function using Design Expert (version 13) was applied to optimize factors for desirable responses. The suggested optimized formulations were prepared and considered as a checkpoint to evaluate the accuracy of the design. The predicted values of each response were determined and compared to their corresponding actual values.

### 3.4. Preparation of Plain NLC and GEF-NLC

The ultrasonic melt-emulsification method was utilized to prepare plain NLC and GEF-NLC as previously reported by Harisa and Badran [48]. Table 1 shows the composition of each formulation based on DOE suggestions. Briefly, the lipid phase was prepared by placing the predetermined amount of solid lipid (SA) and liquid oil (LCFA or LCM or LCT) in a cylindrical beaker to prepare plain NLC. The aqueous phase was prepared by dissolving 200 mg of Pluronic-F68 in 20 mL of distilled water. Both beakers were heated up to 80 °C at the same time. Primary emulsion was obtained following the mixing of the preheated aqueous phase and preheated lipid phase at 5000 rpm. The obtained primary hot microemulsion was subjected to ultrasonication for 3 min at 80% voltage efficiency (10 s of sonication followed by 5 s resting period). For preparing GEF-NLC, 40 mg of pure GEF was added to the lipid phase and the mixture was subsequently subjected to similar preparation steps of plain NLC. The obtained plain NLC and GEF-NLC were instantly placed in the refrigerator for 10 min until cool.

### 3.5. Physicochemical Characterization

#### 3.5.1. PS, PDI, and ZP

A Zetasizer Nano ZS (Malvern Instruments, Malvern, UK) was utilized to measure the physicochemical properties of prepared formulations. Each formulation was diluted in distilled water (1:1000) and evaluated at 25 °C. Dynamic Light Scattering (DLS) and Laser Doppler Velocimetry (LDV) modes were utilized to measure PS, PDI, and ZP. Each value was shown as an average of three independent replicates where each replicate involved six measurements [48].

#### 3.5.2. PXRD

The PXRD spectra of the GEF, processed SA, GEF-NLC_(LCFA)_, and GEF-NLC_(LCT)_ were examined to evaluate the molecular state of SA and GEF crystallinity after preparing GEF-NLC. Moreover, a mixture of lipid core of GEF-NLC_(LCFA)_ and GEF-NLC_(LCT)_ was mixed and melted then cooled to obtain processed SA: LCFA (3:1) and processed SA: LCT (3:1), respectively. This was performed to confirm the effect of liquid oil on SA crystallinity within lipid core of NLC. An X-ray diffractometer (Ultima IV, Rigaku Inc. Tokyo, Japan) was used with a scanning rate of 0.5/min in the scanning range of 3–180°. The characteristic peak of each sample was assessed by collecting the data using monochromatic radiation (Cu Kα’ 1, λ = 1.54 Å), operating at a voltage of 40 kV and current of 40 mA [49].

#### 3.5.3. Drug Content

To determine the amount of drug present in a specific volume of formulation, GEF-NLC was diluted (1:4) in distilled water. In a 10 mL volumetric flask, 1 mL of the formulation was added while the remaining volume was completed with methanol. The obtained dispersion was sonicated for 5 min and centrifuged at 15,000 rpm for 10 min. An aliquot of the supernatant was diluted in acetonitrile (1:10) and subsequently analyzed using the developed UPLC method.

#### 3.5.4. Entrapment Efficiency (EE)

An indirect method was utilized to measure the EE% of the drug in GEF-NLC. Briefly, a predetermined amount of the formulation was centrifuged for 30 min at 80,000 rpm to precipitate GEF-NLC. The amount of the drug in the supernatant was measured using the developed UV-UPLC method. EE% was determined using equation 10 [8,50]:EE = (Total amount of GEF (mg) − Amount of GEF in the supernatant (mg))/(Total amount of GEF (mg)) × 100(10)

### 3.6. In Vitro Release

In vitro release of GEF was performed using a previously described dialysis method with minor modification [51]. The prepared formulation was diluted in phosphate buffer (in a 1:4 ratio) to simulate intestinal conditions. The test was performed by placing the formulation containing 0.5 mg of GEF inside a dialysis membrane bag (molecular weight cut off: 12–14 kDa). This bag was sealed and placed in a beaker containing a preheated 100 mL of simulated intestinal fluid (pH 6.8) containing 0.5% T-80. The beaker was continuously shaken at 100 rpm at 37 ± 1 °C in a thermostat shaker. Samples were withdrawn at 5, 10, 15, 30, 30, 60, 120, 240, 480, 720, 960, and 1440 min and an equal amount of dissolution media was replaced. The withdrawn samples were centrifuged for 10 min at 10,000 rpm and the amount of drug in the supernatant was determined using the developed UV-UPLC method. Formulation performance was compared based on dissolution efficiency (DE)% [31].

### 3.7. Stability Study

During the stability study, the formulations were placed within a 20 mL glass vial with a rubbery stopper. The stability of the prepared plain NLC formulations was evaluated in terms of physicochemical properties upon storage at 4 °C for up to 3 months. Moreover, GEF-NLC_(LCFA)_, and GEF-NLC_(LCT)_ were evaluated in terms of physicochemical properties upon storage at 4 °C at 7-, 15-, 30-, 60-, and 90-day time intervals.

### 3.8. In Vitro Cytotoxicity

Human non-small-cell lung cell line (A549) was obtained from DSMZ Leibniz Institute (German Collection of Microorganisms and Cell Cultures Braunschweig, Germany). It was utilized to study the cytotoxic activity of plain NLC_(LCFA)_, pure GEF, and GEF-NLC_(LCFA)_ [39]. The cytotoxic activity of a drug-free carrier against the A549 cell line was evaluated using plain NLC, where an equivalent volume to GEF-NLC was incubated with cultured cells. The cells were cultured in a DMEM culture medium supplemented with 1% *v*/*v* penicillin–streptomycin and 10% *v*/*v* FBS (Gibco; USA) and maintained in the incubator at 37 °C with 5% CO_2_. Briefly, about 1 × 10^5^ cells were cultured in each well using a 96-well for 24 h before the experiment. The plain NLC, pure GEF, and GEF-NLC were incubated with cultured cells at different concentrations (2.5–20 μg/mL). The effect of each formulation was tested at three time intervals (24, 48, and 72 h) to study its effect over time. At a predetermined interval, 50 μg of MTT was added to the cells and incubated for 4 h in the dark at 37 °C. Next, the formazan product was solubilized with acidified isopropanol and the absorbance was measured at a wavelength of 570 nm using a microplate reader (Bio-Tek, Winooski, VT, USA). IC50 was calculated through the contraction of the dose–response curve. Cell viability (%) was calculated by dividing the optical density of the treated sample by the optical density of the untreated sample and then multiplying by 100.

### 3.9. Statistical Analysis

Physicochemical properties of plain NLC and GEF-NLC were statistically evaluated using SPSS software, Version 26. PS, ZP, IC50, and DE% were compared using an independent t-test (for data with two sets), while a two-way ANOVA test was used to assess the stability of different GEF-NLC formulations upon storage. One-way ANOVA was used to compare IC50 at different time intervals. Data were expressed as mean ± SD. *p*-value < 0.05 was used as the criterion for significance.

## 4. Conclusions

The present study introduced an experimental design and optimization of GEF-NLC for the treatment of metastatic lung cancer. Decreasing the SL/LO ratio significantly reduced PS and PDI. The use of LCFA as a liquid oil led to decreased PS, PDI, and ZP values. In contrast to the burst release of GEF-NLC_(LCT)_ (≈70% within 2 h), GEF-NLC_(LCFA)_ was able to control GEF release (≈57% up to 24 h). In vitro cytotoxicity revealed that GEF-NLC_(LCFA)_ modulates the cytotoxic activity of GEF on A549 cells. Therefore, NLC is a promising strategy to improve the therapeutic impact of GEF in the treatment of lung cancer. GEF-NLC_(LCFA)_ could open new research avenues for improving the GEF therapeutic profile in the treatment of lung cancer with lowering side effects. Further in vivo studies are required to study the bio-distribution of prepared GEF-NLC_(LCFA)_ formulation.

## Figures and Tables

**Figure 1 molecules-28-00448-f001:**
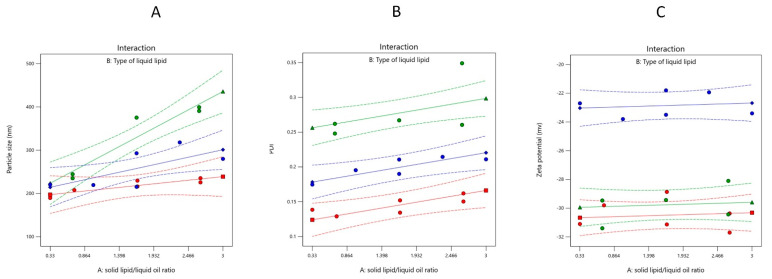
The effect of independent variables, solid lipid: liquid oil ratio and type of liquid oil on the measured responses (**A**) particle size, (**B**) PDI, and (**C**) zeta potential of plain NLC. Red, green, and blue colors represent the curves of a formulation containing LFCA, LCM, and LCT, respectively.

**Figure 2 molecules-28-00448-f002:**
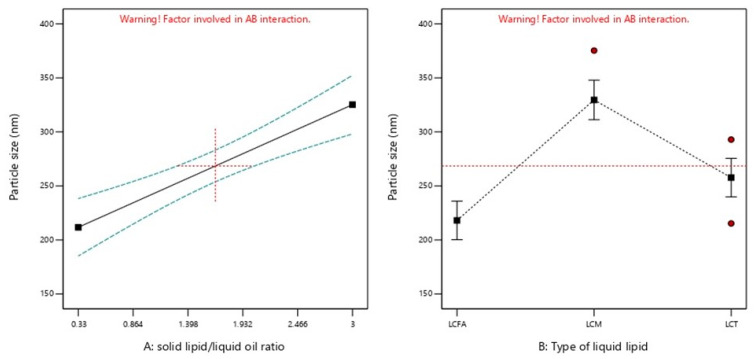
Graphical representation of the effects of (**A**) solid lipid: liquid oil ratio and (**B**) type of liquid oil on droplet size. Red dotted horizontal line and plus sign represent the average value over factor B at the focused point of factor A = 1.665, respectively. Green dotted lines represent 95% CI bands. Red circles and black squares represent design points and predicted values, respectively.

**Figure 3 molecules-28-00448-f003:**
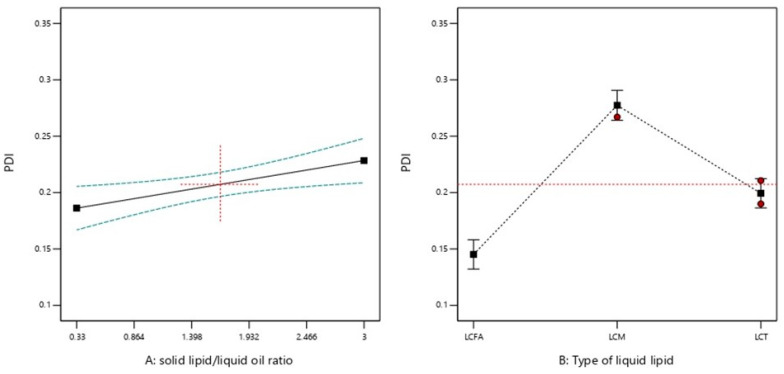
Graphical representation of the effects of (**A**) solid lipid: liquid oil ratio and (**B**) type of liquid oil on PDI. Red dotted horizontal line and plus sign represent the average value over factor B at the focused point of factor A = 1.665, respectively. Green dotted lines represent 95% CI bands. Red circles and black squares represent design points and predicted values, respectively.

**Figure 4 molecules-28-00448-f004:**
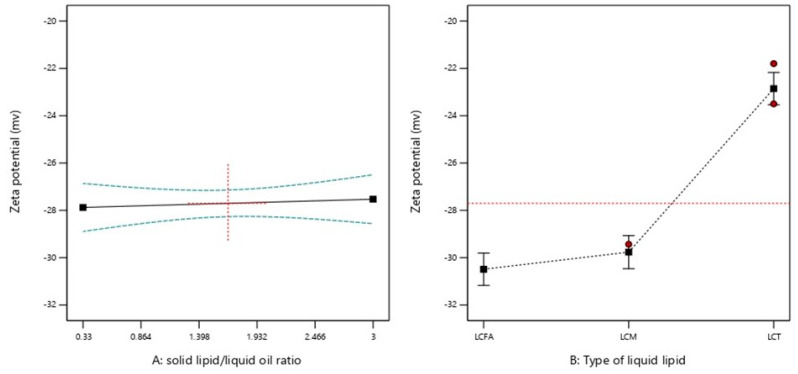
Graphical representation of the effects of (**A**) solid lipid: liquid oil ratio and (**B**) type of liquid oil on zeta-potential value. Red dotted horizontal line and plus sign represent the average value over factor B at the focused point of factor A = 1.665, respectively. Green dotted lines represent 95% CI bands. Red circles and black squares represent design points and predicted values, respectively.

**Figure 5 molecules-28-00448-f005:**
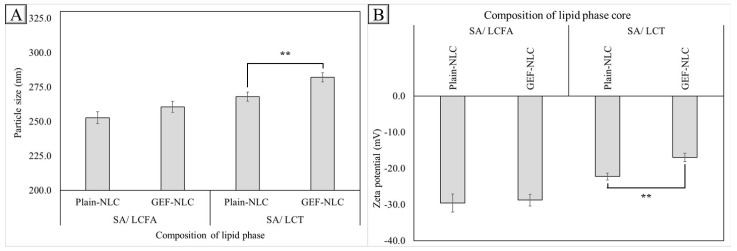
Influence of GEF loading on (**A**) PS and (**B**) ZP of different plain NLC formulations. Data were expressed as the mean ± SD, N = 3, *p*-value significant at ** 0.01.

**Figure 6 molecules-28-00448-f006:**
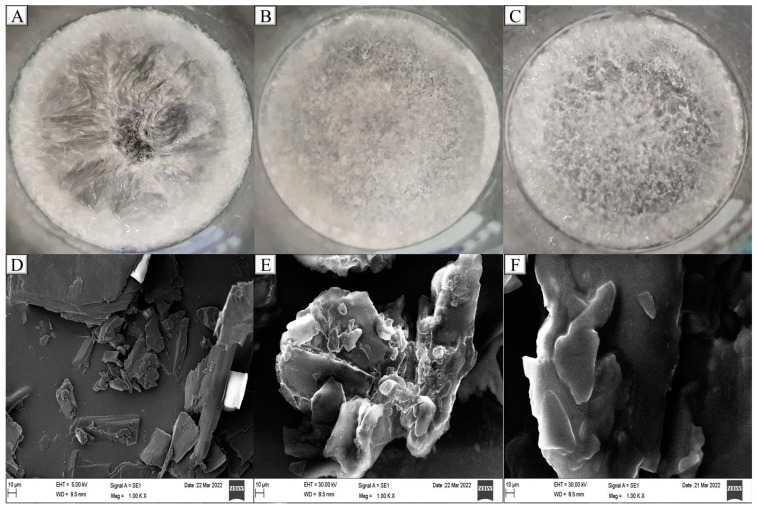
Morphological appearance and SEM image of crushed crystals of (**A**,**D**) SA, (**B**,**E**) SA: LCFA (3:1), and (**C**,**F**) SA:LCT (3:1). The agents were physically mixed and heated to obtain a melted lipid mixture and then cooled in the refrigerator.

**Figure 7 molecules-28-00448-f007:**
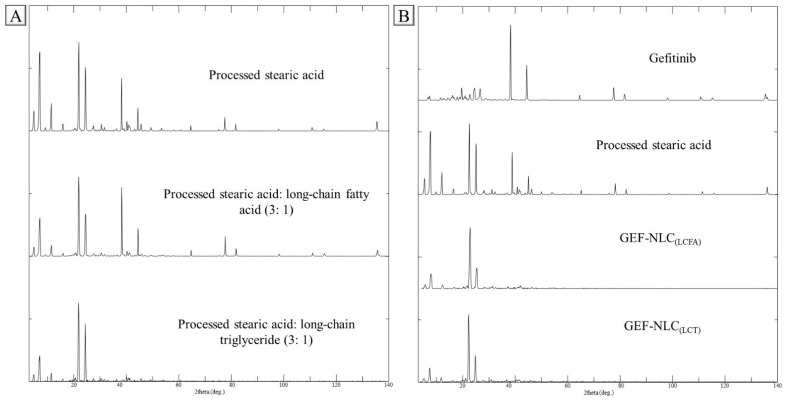
PXRD of (**A**) processed SA, processed SA:LCFA (3:1), and processed SA:LCT (3:1), (**B**) GEF, processed SA, GEF-NLC_(LCFA)_, and GEF-NLC_(LCT)_.

**Figure 8 molecules-28-00448-f008:**
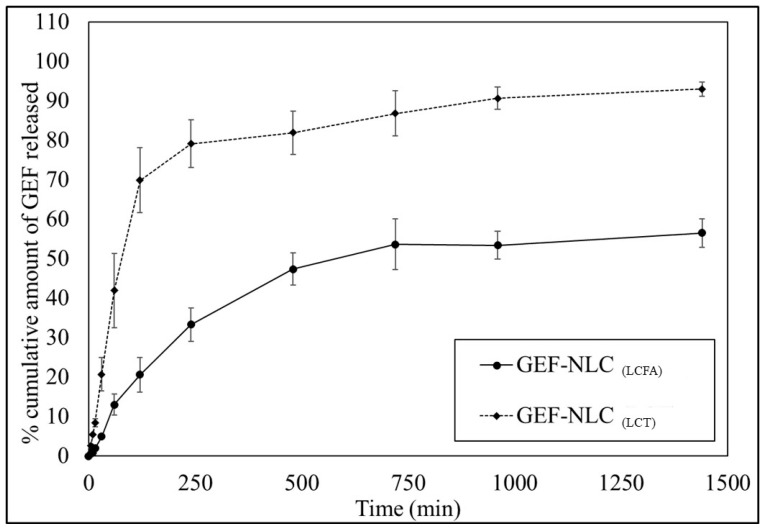
In vitro release of GEF from GEF-NLC_(LCFA)_ and GEF-NLC_(LCT)_ formulations. Data are expressed as the mean ± SD, N = 3.

**Figure 9 molecules-28-00448-f009:**
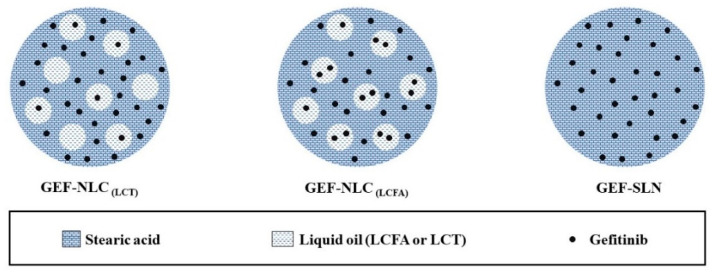
Schematic diagram of lipid core distribution of SA, LCT, LCFA, and GEF within GEF-NLC_(LCT)_, GEF-NLC_(LCFA)_, and GEF-SLN. LCT showed limited GEF solubility compared to LCFA.

**Figure 10 molecules-28-00448-f010:**
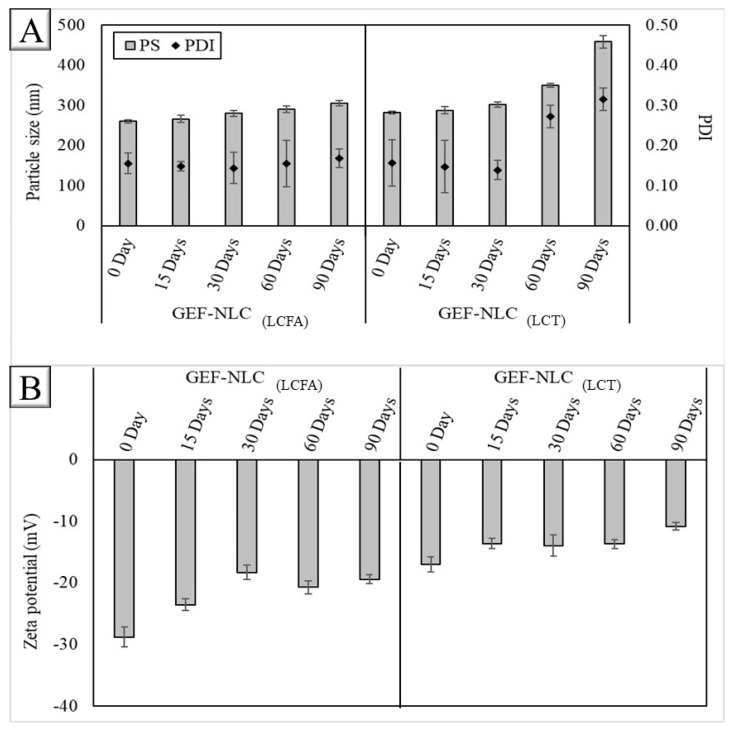
(**A**) Particle size and PDI and (**B**) zeta potential values of GEF-NLC_(LCFA)_ and GEF-NLC_(LCT)_. Data are expressed as the mean ± SD, N = 3.

**Figure 11 molecules-28-00448-f011:**
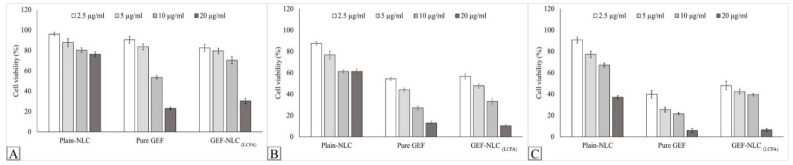
Effect of plain NLC, pure GEF, and GEF-NLC_(LCFA)_ on A549 cell viability after (**A**) 24 h, (**B**) 48 h, and (**C**) 72 h, respectively. Data are expressed as mean ± SD, N = 3.

**Figure 12 molecules-28-00448-f012:**
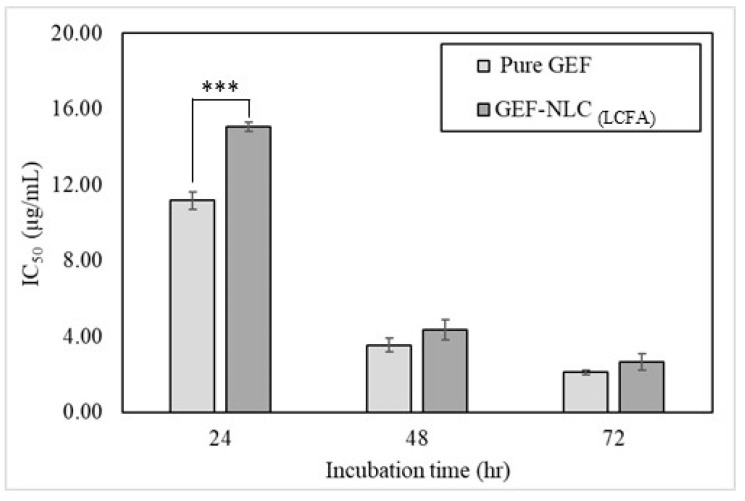
IC_50_ of pure GEF and GEF-NLC_(LCFA)_ following 24, 48, and 72 h incubation. Data are expressed as mean ± SD, N = 3, *p*-value significant at *** 0.001.

**Table 1 molecules-28-00448-t001:** Physicochemical properties of suggested plain NLC formulation based on the DOE model.

Formulation Code	Factors		Responses			
	X_1_: Solid Lipid: Liquid Lipid Ratio (SL/LO)	X_2_: Type of Liquid Lipid	Y_1_: PS (nm)	Y_2_: PDI	Y_3_: ZP (mv)	Y_4_: Aggregation upon Storage
9	0.33	LCFA	189.6	0.139	−31.1	Yes
8	0.70	LCFA	207.7	0.129	−29.8	Yes
5	1.68	LCFA	216.1	0.134	−31.1	No
12	1.68	LCFA	229.6	0.152	−28.9	No
4	2.65	LCFA	235.3	0.162	−31.7	No
7	2.65	LCFA	225.6	0.15	−30.4	No
1	0.68	LCM	235.1	0.262	−31.4	Yes
14	0.68	LCM	244.5	0.248	−29.5	Yes
17	1.67	LCM	375.3	0.267	−29.4	Yes
3	2.63	LCM	390.6	0.261	−30.4	Yes
6	2.63	LCM	399.1	0.349	−28.1	Yes
13	0.33	LCT	220.7	0.175	−22.7	No
10	1.00	LCT	219.5	0.195	−23.8	No
15	1.67	LCT	292.9	0.211	−23.5	No
16	1.67	LCT	215.2	0.19	−21.8	No
11	2.33	LCT	318	0.215	−21.9	No
2	3.00	LCT	279.9	0.211	−23.4	No

LCFA: long-chain fatty acid, LCM: long-chain monoglyceride, LCT: long-chain triglyceride, PS: particle size, PDI: polydispersity index, ZP: zeta potential.

**Table 2 molecules-28-00448-t002:** ANOVA analysis of the measured responses for the selected models.

Response	Selected Model	Degree of Freedom	Adjusted R^2^	Predicted R^2^	F-Value	*p*-Value
PS	2FI	2	0.8346	0.7771	5.97	0.0175
PDI	Linear	3	0.8838	0.8246	35.91	<0.0001
ZP	Linear	3	0.9184	0.8868	61.03	<0.0001

PS: particle size, PDI: polydispersity index, ZP: zeta potential.

**Table 3 molecules-28-00448-t003:** ANOVA of the quadratic model presenting the correlation (*p*-value) between independent formulation variables and measured physicochemical properties (PS, PDI, and ZP).

Response	X1: *p*-Value of Solid Lipid: Liquid Oil Ratio	X2: *p*-Value of Type of Liquid Lipid
PS	0.0003	0.0001
PDI	0.0151	<0.0001
ZP	0.6680	<0.0001

PS: particle size, PDI: polydispersity index, ZP: zeta potential.

**Table 4 molecules-28-00448-t004:** Validation of the experimental design model (SL/LO = 3, Plain-NLC_(LCFA)_).

Response	n	SD	Predicted Mean	SE Pred	95% PI Low	Data Mean	95% PI High
PS	3	27.40	238.89	26.2	181.3	252.7	296.5
PDI	3	0.023	0.162634	0.018	0.123	0.183	0.202
ZP	3	1.068	−30.314	0.86	−32.2	−29.6	−28.5

PS: particle size, PDI: polydispersity index, ZP: zeta potential.

**Table 5 molecules-28-00448-t005:** Validation of the experimental design model (SL/LO = 3, Plain-NLC_(LCT)_).

Response	n	SD	Predicted Mean	SE Pred	95% PI Low	Data Mean	95% PI High
PS	3	27.40	301.1	26.0	243.8	268.1	358.3
PDI	3	0.023	0.220	0.016	0.185	0.193	0.256
ZP	3	1.068	−22.68	0.85	−24.5	−22.3	−20.8

PS: particle size, PDI: polydispersity index, ZP: zeta potential.

**Table 6 molecules-28-00448-t006:** Drug content and EE% of GEF-NLC_(LCFA)_ and GEF-NLC_(LCT)_.

Formulation	Drug Content (mg/mL)	EE (%)
GEF-NLC_(LCFA)_	2.13 ± 0.48	94.48 ± 2.14
GEF-NLC_(LCT)_	2.09 ± 0.75	91.94 ± 3.08

## Data Availability

Not applicable.

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
