# Peer review of "Optimization of Gefitinib-Loaded Nanostructured Lipid Carrier as a Biomedical Tool in the Treatment of Metastatic Lung Cancer"

_molecules, 2023, doi:10.3390/molecules28010448_

Round 1

Reviewer 1 Report

Optimization of Gefitinib Loaded Nanostructured Lipid Carrier for Biomedical Application in the Treatment of Metastatic Lung Cancer

In the manuscript entitled " Optimization of Gefitinib Loaded Nanostructured Lipid Carrier for Biomedical Application in the Treatment of Metastatic Lung Cancer” by Abdelrahman Y. Sherif et al. the authors report the development of nanostructured lipid carriers for the incorporation of Gefitinib (GEF). The aim was to promote a sustained release of GEF from its delivery system in vivo, to reduce GEF toxicity while providing a lymphatic targeted delivery of the anticancer drug. The authors first developed plain-NLC formulations using Design of Experiments (DOE) approach. The optimized NLC was then loaded with GEF and the final delivery system was characterized using several physicochemical parameters. The cytotoxicity of the GEF-NLC formulation was evaluate in vitro on A549 cells as a model for lung cancer. It is an interesting and good quality research work. 

From my point of view, the results and discussion sections should be joined together. This way some repetitions would be eliminated, and the manuscript results can be discussed and compared with previous works in an easier manner. Another aspect that needs major revision is the DOE part of the work. In section 2.5 the authors need to explain why they seem to prefer NLC with higher particle size (close to 300 nm) in detriment of particles with a mean size closer to 200 nm. It is not clear, when comparing this section with results presented in Figure 1 why the optimized formulation involved using a SL/LO =3, when the aim is to minimize the responses (PS, PDI and ZP). This is true for both plain-NLC (with LCFA or LCT). In the discussion section 3.2 the authors seem to prefer again smaller particles. All these results and conclusions appear to be in contradiction. Authors need to justify the choice of high SL/LO ratio to prepare the GEF-NLC.

Other minor aspects of the manuscript that should be improved:

A general revision of the document is needed to eliminate minor errors. In the introduction section there is an excessive use of adverbs in the beginning of the sentences. Please revise.

Line 40: Please replace “lower” by “low”, and “GEF oral bioavailability ≈44%” by “GEF as an oral bioavailability of around 44%”.

Line 46: Please rephrase the sentence. The term “augments” is not the most correct when referring to the role of nanocarriers in the in vivo biodistribution profile of a drug.

Table 1. Please replace the term “Type of liquid lipid” with “Type of liquid oil” to maintain uniformity of terms and abbreviations.

Line 170: Please replace “LCFA” by “LCT”

Line 283: Please replace “decays” by “decades”.

Line 504: The abbreviation for the liquid oils (OA, ML and SB) were not introduced in the manuscript. Please revise and refer the actual name of the liquid oils used in this study.

Figure 11: The concentration of the GEF used in this assay is not mentioned. The conclusions drawn from this assay are valid for all GEF tested (presented in Figure 10A)?

Author Response

Dear Reviewer,

Thank you for careful reading our manuscript and for raising constructive comments which significantly helped us to improve the manuscript. All the comments have been addressed and the manuscript has been revised accordingly. 

For a point-to-point response to all the comments and corresponding modifications, please see the attachment. 

Best,

Reviewer 2 Report

The authors conducted an extensive study to optimize the composition of nanostructured lipid carriers (NLC) of the drug Gefitinib (GEF), a tyrosine kinase inhibitor (TKI), as a lymphatic drug delivery system for the treatment of metastatic lung cancer. They chose to compare three liquid lipids as the main carriers along with solid phase lipids and examined a range of solid:liquid lipid ratios. For the plain carrier, they measured particle sizes, polydispersity, zeta potential, drug loading capacity, entrapment efficiency and stability during long term storage. For GEF-loaded carriers they measured drug solubility and release rates and compared cytotoxicity between GEF alone and GEF-NLC carriers.

This is a good study pointing to a potentially beneficial formulation for GEF drug delivery with characteristics that promise better targeting and fewer side effects in the treatment of lung cancer.

It is recommended that it be accepted for publication. There are a few issues that are not addressed clearly, and the acronym-rich presentation makes the manuscript difficult to follow.

Here are a few issues/suggestions:

What is the solid lipid (SA) used in the different formulations. Presumably, it is Stearic Acid but that is not spelled out until much later in the manuscript. Suggestion that it is made clear at the very beginning that SL used throughout is SA, and then use only one or the other after that.

Some of the acronyms are never spelled out.

The text is already very heavy on acronyms making it tough to follow!

It would be helpful to create table with all the acronyms.

The organization of Table 1. is hard to follow. Ordering according to “Formulation code” is not helpful and not sure what the “Formulation Code” is. Maybe ordering according to the liquid lipid type, followed by SL/LO ratio would make things easier.

What is DF in Table 2?

Fig.10 includes the result of cell viability experiment using plain-NLC. However, plain-NLC contain no GEF – right? If so, why are there four bars for different concentrations of the drug that is not there? And what does the drop in viability mean with increased drug concentration?

Some extra small-print text in the figures is not explained and maybe redundant and better be removed.

The section on the selection of formulation is lacking adequate explanation. Does everybody know what “Factor A and B” mean?

Concerning GEF-NLC stability, was it ascertained that GEF was not released during lengthy storage? Was the cytotoxic effect still there after the 90 day storage?

Author Response

Dear Reviewer,

Thank you for careful reading our manuscript and for raising constructive comments which significantly helped us to improve the manuscript. All the comments have been addressed and the manuscript has been revised accordingly. 

For a point-to-point response to all the comments and corresponding modifications, please see the attachment. 

Best wishes,,,

Round 2

Reviewer 1 Report

The authors made a great effort to revise the manuscript according to the reviewer comments.

Accepted for publication